# A Mixed Methods Protocol for Developing Strategies to Improve Access to Health Care Services for Refugees and Asylum Seekers in Gauteng Province, South Africa

**DOI:** 10.3390/healthcare11172387

**Published:** 2023-08-24

**Authors:** Duppy Manyuma, Takalani Grace Tshitangano, Azwinndini Gladys Mudau

**Affiliations:** Department of Public Health, University of Venda, Private Bag X 5050, Thohoyandou 0950, South Africa

**Keywords:** access, asylum seekers, refugees, strategies

## Abstract

Access to healthcare is a fundamental human right for all people, including refugees and asylum seekers. Despite the provision in the Refugee Act of South Africa, which allows refugees to enjoy the same access to health services as the citizens of the Republic, refugees still encounter challenges in accessing such services in Gauteng province. This protocol aims to develop strategies to improve access to health care services for refugees and asylum seekers in Gauteng province, South Africa. The study will be conducted in three phases. An exploratory sequential mixed methods design will be employed in phase 1 of the study; the initial study will be based on a qualitative approach followed by a quantitative approach. Phase 1 (a) of this study will employ a qualitative approach in Gauteng province among conveniently sampled health practitioners as well as purposively sampled refugees and asylum seekers. Interviews will be used to collect data that will be analyzed thematically. Phase 1 (b) will adopt a quantitative approach based on the findings from the initial qualitative study. The ethical principles of informed consent, anonymity, privacy, confidentiality, and avoidance of harm will be adhered to throughout the research process. Phase 1 (c) will be meta-inference and conceptualization. Phase 2 will focus on the development of strategies using strength, weakness, opportunities, and threats analysis and a build, overcome, explore, and minimize model to guide the process. In Phase 3, the Delphi technique will be used to validate the developed strategies. The conclusion and recommendations will be based on the findings of the study.

## 1. Introduction

The degree to which non-citizens, especially asylum seekers, should be accommodated in the provision of social services such as health care is one of the key concerns the state has had to confront with the development of international migration [1]. The 1951 United Nations High Commissioner for Refugees (UNHCR) convention defines a refugee as “any person who has a well-founded fear of being persecuted for reasons of race, religion, nationality, membership of a particular social group or political opinion, is outside the country of their nationality and is unable or, owing to such fear, is unwilling to avail him/herself of the protection of that country” [2] (p. 152). An asylum seeker is defined as a person who is in the application process of being granted refugee status by the host country [3].

According to United Nations High Commissioner for Refugees [4], approximately 108.4 million displaced people were recorded globally by the end of 2022. From this recorded number, 62.5 million were internally displaced people, 35.3 million were refugees, and 5.4 million were asylum seekers.

The UN General Assembly held a high-level conference on the 19 of September 2016 to discuss the influx of refugees and migrants and develop a system to respond to the large movement of refugees and migrants [5]. During the September 2016 UN Summit, 193 member states signed up for the New York Declaration [6]. The declaration was to affirm the member states’ commitment to tackle the issue of HIV and other health care needs faced by migrants, refugees, and asylum seekers, as well as ensuring smooth usage of HIV prevention, treatment, care, and support services for migrant communities [6,7].

According to Robertshaw, Dheshi, and Jones [3], refugees and asylum seekers are considered to be vulnerable groups with various health needs, since most of them originate from countries with a high prevalence of pre-existing infectious diseases such as human immunodeficiency virus (HIV), tuberculosis (TB), and hepatitis B. Chauh et al. [8] believe that the journeys that refugees and asylum seekers take from their home countries to host countries can expose them to various diseases. Furthermore, the receiving countries are responsible for attending to the health needs of those that they are hosting. Campbel et al. [9] state that the health needs of refugees and asylum seekers should be well understood to improve their health. Newbold [10] and Levesque et al. [11] are of the view that improved health is the necessary individual means for refugees and asylum seekers to reach their full potential and to successfully integrate into the labor market, society, and culture of the host country/community.

### 1.1. Definition of Concepts

Refugees

The term “refugee” refers to a person who has a sincere fear of persecution on the grounds of racial, religious, national, social, or political origin, who is outside of the country of which they are a citizen and is unable, or unwilling, to return to their country of origin [2]. In this study, the term “refugee” refers to a person in possession of a Section 24 permit or refugee ID who is suffering from an infectious/non-communicable disease.

Asylum seekers

The term “asylum seeker” refers to a person who has applied for or is in the process of applying for asylum in a host country [3]. In this study, the term “asylum seeker” refers to a person in passion of a valid or expired Section 22 permit who is suffering from an infectious/non-communicable disease.

Access

The Institute for Medicine [12] defines “access” as the timely use of personal health services to achieve the best health outcomes. In this study, “access” refers to affordability, availability, accessibility, acceptability, and accommodation of the health care system to refugees and asylum seekers for them to acquire health information, screening, and treatment for infectious/non-communicable diseases at any time when in need of such services without any struggle.

### 1.2. Mechanisms to Improve Access to Health Care for Refugees

In Austria, refugees and asylum seekers have access to free health care services. Once a person applies for asylum in Austria, they are granted health insurance as part of the basic services. The medical assistance covers access to public hospitals, psychological treatment, and medications. Asylum seekers whose applications have been positively reviewed are then recognized as refugees and granted an e-card (electronic health certificate), which is required by those consulting and those seeking treatment in the country. The e-card allows refugees to be in the same health schemes as Austrian citizens, and they can also apply for an exemption if their monthly income is under a specific amount; on the other hand, asylum seekers are not liable for paying the small prescription fees for medical drugs [13].

Malaysia is one of the countries known for successfully improving the health of its community through low-cost, universal, and comprehensive service provision. However, it was found that refugees access health services at a higher price than the citizens of Malaysia. In 2005, UNHCR in Malaysia and the Ministry of Health signed a memorandum of understanding, which stated that all refugees who are recognized by UNHCR will be given a 50% discount on their medical costs [14]. However, Chuah et al. [15] states that the cost of accessing health services for refugees in Malaysia is still high, despite them being granted a 50% discount. In 2017, the UNHCR in Malaysia, together with the RHB Bank introduced refugee medical insurance, which aimed at improving access to health care services [16]. However, the enrolment of UNHCR cardholders was recorded at only 12.2%. The low registration of refugees to the medical insurance scheme led to its closure in June 2018, since it was becoming expensive for the partnering bank [17]. A healthcare financing system was also launched in Malaysia called the “Hospitalisation and Surgical Scheme for Foreign Workers” (SPIKPA), where each employed migrant makes an annual contribution of MYR 120, which allows them to enjoy a cover of MYR 20,000 per annum. The annual cover provided by SPIKPA is still considered to be small, since non-citizens are subjected to high charges at public health facilities [18].

According to Brandenberger et al. [19], in 2002, the European Union initiated a program called the Migrant-Friendly Hospital Project to improve the delivery of health care services to migrant populations. The focus of the project was on increasing interpreting services and providing health information that was relevant to the health needs of migrants in a manner appropriate to their cultural beliefs. This was achieved through the training of health professionals in cultural competencies. According to Pusztai et al. [20], in Serbia, the United Nations country team organized resources from the European delegation to strengthen the country’s health system to be able to respond to the health needs of migrants and refugees and to improve access to health services. Part of the strategy to improve Serbia’s health system was the procurement of equipment and improving the infrastructure for medical facilities within municipalities receiving a high volume of refugees, advancing technical regulations, and training health professionals, health specialists, and cultural mediators in raising awareness activities for refugees. In England, all persons are entitled to free primary care, general practitioner access, and nurse consultations despite their migration or economic status. All persons are further entitled to the following services, which are also free: accident and emergency, family planning, and the diagnosis and treatment of communicable diseases and sexually transmitted infections, as well as treatment of conditions resulting from torture, trafficking, domestic violence, and female genital mutilation. Refugees, asylum seekers, and undocumented migrants do not have the same rights when it comes to secondary healthcare in England, their rights depend on their migration status at that particular time. Accepted refugees and asylum seekers, as well as those who were rejected but have appealed the decision, enjoy free access to treatment, whereas those with rejected status or are undocumented have to pay for secondary health care [21].

In Musina, The Department of Health had a mobile clinic that was meant to service migrants working on the farms; however, the mobile clinic was found to be under-resourced and unable to provide HIV care and support to the migrant communities [22]. In response to the HIV care and support needs of migrants in Musina, MSF introduced the Musina Model of Care, which included mobile clinic programs to render voluntary counseling and testing (VCT) for HIV and antiretroviral therapy (ART). A team of community health workers (CHWs) was also recruited to work alongside the mobile clinic [23]. Furthermore, during the time that MSF was implementing its programs, the International Organization for Migration (IOM), together with other local non-governmental organizations, trained a total of 103 peer educators as part of their Ripfumelo project, which aimed at reducing the rate of HIV and TB amongst refugees, migrants and persons on the move [23]. Brandenberger [19] believes that it is important to identify and supply the resources needed for the health care system to be able to respond to the health needs of refugees and asylum seekers.

South Africa (SA) has a constitutional mandate to provide quality healthcare services to all who are living within the borders of the country [24]. Refugees and asylum seekers residing in South Africa also have the right to access quality health care services [25]. These rights are enshrined in Section 27 (g) in the Refugee Act of 1998, which clearly states that “a refugee is entitled to the same basic health services and basic primary education which the inhabitants of the Republic receive from time to time” [26] (p. 20).

The South African health care system has been reorganized, just like in other developing countries, to improve access and the quality of service delivery to vulnerable populations. The health system has been structured into three levels, which are: national, provincial, and local government [24]. It is believed that decentralization of the health care system helps to improve the quality of services in developing countries [27]. The South African health system is reported to have been able to reduce the inequality gap that existed in terms of health care access post-apartheid in 1994 [28].

South Africa is reported to be one of the countries responding well to the treatment and care of people living with HIV and TB [29]. SA initiated the Central Chronic Medicines Dispensing and Distribution (CCMDD) program, which sought to improve access to medicines for stable patients on ARVs, patients on ARVs with comorbidities, and those patients with non-communicable diseases (NCDs) needing chronic therapy [30]. The CCMDD program was first rolled out in 10 districts in 2014; however, seeing the good results it produced in the initial districts, it was then extended to 46 districts across eight provinces (the Western Cape province did not benefit from this program since it had a similar project running), reaching out to a total of 1.7 million persons [30,31].

With an increase in the number of people who can access treatment for HIV and TB, adherence is gradually becoming a problem [31]. The National Department of Health established the National Adherence Guidelines for Chronic Diseases, including HIV, TB, and NCDs, to improve treatment adherence, working towards Vision 2030 of the National Development Plan. The overall goal of the guidelines is to enhance access to quality and relevant services for persons with chronic illnesses and to ensure that there is adherence [32].

## 2. Materials and Methods

### 2.1. Aim and Objectives

The protocol is for a study that aims to develop strategies to improve access to health care services for refugees and asylum seekers in Gauteng province, South Africa.

The protocol objectives are described according to the phases in which the study will be conducted. The objectives are:


Phase 1 (a) (Qualitative approach)


To explore the experiences of refugees and asylum seekers regarding access to health care services in Tshwane and Johannesburg Metropolitan Municipalities, Gauteng province;To describe the challenges faced by health practitioners when rendering health services to refugees and asylum seekers in Tshwane and Johannesburg Metropolitan Municipalities, Gauteng province;To describe the support needed to improve access to health care services for refugees and asylum seekers in Tshwane and Johannesburg Metropolitan Municipalities, Gauteng province.


Phase 1 (b) (Quantitative approach)


Objectives will be based on the findings of phase 1 (a) (Qualitative approach).


Phase 1 (c)


To draw overall conclusions based on the findings from the qualitative and quantitative approaches.


Phase 2


To develop strategies to improve access to health care services for refugees and asylum seekers in Gauteng province.


Phase 3


To validate the developed strategies to improve access to health care services for refugees and asylum seekers.

### 2.2. Research Question

How can access to health care services be improved among refugees and asylum seekers in Gauteng province, South Africa?

### 2.3. Theoretical Framework

This study will be conceptualized within Penchansky and Thomas’ theory of access. Penchansky and Thomas [33] believe that access in the health context is complex and should be understood within the 5 elements/characteristics, namely: affordability, availability, accessibility, acceptability, and accommodation. Significantly, no element should be looked at to improve health care since they all play a critical role in fostering access to services [34]. Andersen [35] argues that the theory of access is best suitable for studies focusing on the experiences of patients about accessing health services. The researcher found the theory to be relevant to this study since some of its objectives will be focused on exploring the experiences of refugees and asylum seekers regarding access to health care services.

Affordability

Affordability is influenced by the fees charged by the service providers and the client’s willingness or means to pay for such services. It further ensures that there is value for money, the services being rendered should be worth the amount of money that the clients are being charged. The Refugee Act states that all refugees should be treated as South Africans when accessing health services, which means that a means test should be applied to determine the amount that a refugee should be charged at health facilities;

Availability

Availability assesses the level at which the service provider can meet the needs of the clients using the relevant resources, such as technology and personnel with the required skills and knowledge about the nature of services being rendered;

Accessibility

Accessibility is controlled or guided by the extent to which a client can physically reach the service provider’s facility. This also includes the distance that one has to travel to obtain the services and the environment’s friendliness in terms of being able to accommodate clients with disability;

Accommodation

The accommodation reflects the level at which the organization’s set-up or operation is structured to meet the need of the clients with ease. It looks at the possibility of clients obtaining the required services without prior appointments, the amount of time that they need to wait before they can be attended to, and how telephone communications are arranged with that particular organization;

Acceptability

Acceptability focuses on the level at which the client is comfortable receiving services or the service provider’s extent of comfort in rendering the services based on one’s unchangeable characteristics. The characteristics include the age, sex, social class, and ethnicity of the client or the service provider.

Penchansky and Thomas’ theory of access is considered to be relevant in these studies since it focuses on different components or dimensions to improve access to health care services. It is believed that the use of this theory will contribute to the development of well-balanced strategies to improve access to health care services for refugees and asylum seekers.

In this study, the constructs of Penchansky and Thomas’ theory of access will be used to guide data collection tools, data analysis, and SWOT analysis, as well as in the BOEM development of strategies.

### 2.4. Research Methodology

The study will be conducted in various phases. Phase 1 will be a needs assessment with three stages. Phase 1 (a) of this study will employ a qualitative approach with exploratory and descriptive designs. Phase 1 (b) will adopt a quantitative approach based on the results obtained from the qualitative approach. Phase 1 (c) will be meta-inference and conceptualization. Phase 2 will be the development of strategies to improve access to health care services for refugees and asylum seekers guided by the strength, weakness, opportunities, and threats (SWOT) analysis and build, overcome, explore, and minimize (BOEM) development. Phase 3 will focus on the validation of the developed strategies using the Delphi technique.

An exploratory sequential mixed methods design will be employed in this study. The study will first explore the phenomenon under investigation using a qualitative approach. After collecting and analyzing the qualitative data, themes will be generated and used to inform the development of a quantitative instrument that will be used to further explore the research problem. The exploratory sequential mixed method will further assist the researcher in deciding which variables need to be measured during the quantitative study. The use of this mixed method is considered to be relevant for this study since it will assist the researcher in clarifying, elaborating, enhancing, and illustrating the results from one method using the other.

### 2.5. Phase 1 (a): Qualitative Approach

A qualitative approach will foster direct interaction between the researcher and the participants, leading to an in-depth understanding of refugees’ and asylum seekers’ experiences regarding access to health care services and the challenges faced by health practitioners when rendering these services. This approach will award participants the opportunity to narrate their lived experiences and challenges without any limitations [36].

#### 2.5.1. Research Design


Exploratory design


The study will explore the experiences of refugees and asylum seekers regarding access to health care services, as well as the challenges faced by health practitioners when providing health services to refugees and asylum seekers. The researcher will further explore the kind of support needed to improve access to health care services for refugees and asylum seekers [37].


Descriptive design


Refugees and asylum seekers will be given an opportunity to describe their experiences regarding access to health care services, whereas health practitioners will be awarded the opportunity to describe the challenges that they encounter when providing health services to refugees and asylum seekers. Strategies to improve access to health care services for refugees and asylum seekers will also be described [38].

#### 2.5.2. Study Setting

The study will be conducted in Gauteng province in South Africa. Gauteng is the smallest province in South Africa, yet it is considered to be the richest and overcrowded. It consists of three metropolitan municipalities (the City of Ekurhuleni, the City of Johannesburg, and the City of Tshwane Metropolitan Municipality) and two district municipalities (Sedibeng District and West Rand District Municipality).

Gauteng is considered to be the economic hub of South Africa and attracts refugees, asylum seekers, economic migrants, and domestic migrants from rural areas such as Limpopo, KwaZulu Natal, and Eastern Cape. Gauteng province is a preferred destination for many refugees, asylum seekers, and economic migrants because of better economic opportunities, jobs, and the promise of a better life.

According to the Department of Statistics South Africa [39], Gauteng is regarded as the province with the largest share of the South African population. With an estimated number of 15.5 million people living in the province, this number constitutes 26% of the total South African population. Furthermore, the Department of Statistics South Africa indicates that Gauteng hosts approximately 1,553,162 migrants.

#### 2.5.3. Population and Sampling

The study population will consist of medical doctors and nurses who are working in public hospitals in Gauteng province, as well as refugees and asylum seekers who are residing in Gauteng province who make use of public health care services.


Sampling


Sampling will be performed in a multiple stages. There will be sampling of metropolitan/district municipalities, sampling of hospitals, sampling of health practitioners, and sampling of refugees and asylum seekers.

Sampling of Metropolitan/District Municipalities

Two metropolitan municipalities will be purposively sampled for the study. The sampling of the metropolitan municipalities is based on the researcher’s judgment and knowledge of the areas where most refugees and asylum seekers stay in Gauteng. The researcher has worked in Gauteng on a project involving refugees and asylum seekers, his experience will contribute to the purposive selection of the metropolitan municipalities. The researcher will sample the City of Tshwane and the City of Johannesburg Metropolitan Municipality;

Sampling of Hospitals

In this study, hospitals will be sampled purposively. The researcher will use his judgment to select hospitals that are closer to areas where refugees and asylum seekers reside within the two selected metropolitan municipalities. Four hospitals (Kalafong Hospital, Steve Biko Academic Hospital, Charlotte Maxeke Academic Hospital, and Helen Joseph Hospital) will be sampled for this study. The four selected hospitals are closer to areas where refugees and asylum seekers stay. Based on the researchers’ experiences when working in Gauteng, the four hospitals were mostly preferred by refugees and asylum seekers;

Sampling of Health Practitioners

Convenience sampling will be used to select health practitioners rendering health care services to refugees and asylum seekers at the sampled health facilities. Health practitioners who will be present during the time of the visit to the health facility will be conveniently sampled. A total of 80 health practitioners will be selected to form part of the study. Twenty health practitioners will be selected to participate in the study from each of the four sampled hospitals. However, the actual number of health practitioners to be included in the study will be determined by data saturation;

Sampling of Refugees and asylum seekers

After getting permission from the Department of Health, the researcher will use the records at the selected hospitals to sample participants who have sought health care services. Those who appear to have the information that is needed by the researcher will be recruited by telephone. They will be informed about the study and, if they agree to be part of it, the researcher will make further arrangements with them for data collection. A total of 40 refugees and asylum seekers will be selected to participate in the study. However, the actual number of refugees and asylum seekers who will be interviewed will be guided by data saturation;

Sample size

In this study, 40 refugees and asylum seekers will be interviewed, of which, 20 will be from Pretoria and 20 will be from Johannesburg. Furthermore, a total of 80 health practitioners will be interviewed for this study. Only 20 health practitioners in each of the four selected health institutions will be interviewed. However, the actual number will be determined by data saturation.

#### 2.5.4. Data Collection

An interview guide with semi-structured questions will be used in this study. The interview guide consists of two questions in each category to collect data regarding access to health care services for refugees and asylum seekers.


Preparation of Participants


The researcher will collect the details of refugees and asylum seekers who have sought health care services from the health facilities’ records and recruit them to participate in the study, as they are considered to be the relevant participants. Appointments for data collection will be set based on the times and dates suitable to the participants. Participants will also have the freedom to choose the place where the interviews should be conducted; this method was chosen to make the participants feel comfortable.

Health practitioners working in any of the selected health facilities, who have rendered health services to refugees and asylum seekers, will be visited at the health facilities for recruitment to participate in the study. Appointments will be set with those willing to be part of the study for data collection based on a time suitable to them. The interviews will be conducted at the health facilities.

All participants will be given full details about the study to help them to decide if they would like to be part of the study or not. Those who are willing to participate in the study will be requested to give verbal or written consent before the interviews. The participants will be informed that their participation is voluntary and if they wish to withdraw from the study there would be no consequences. The researcher will reassure all the participants that their details will not be used when reporting the study and that the information they provide will only be shared with those directly involved in the study.

The participants will be requested to give permission for the interviews to be recorded. The purpose of recording the interviews will be well explained to the participants, and if they do not want certain information to be recorded the researcher will immediately stop the recording and continue recording on the instruction of the participants.Refugees and asylum seekers

Individual telephone interviews will be conducted to collect data from the refugees and asylum seekers who will be sampled to form part of the study. The interviews will be directed by the following questions:As a refugee or asylum seeker who has once sought health services, kindly share with me your experience regarding access to such services;What do you think should be done to improve your access to health care services?


Health practitioners


Data will be collected through the use of individual in-depth interviews between the researcher and health practitioners who have experience in providing health care services to refugees and asylum seekers. The interviews will be guided by the following questions:As a health practitioner who has once rendered health care services to a refugee or an asylum seeker, can you kindly share with me some of the challenges you encountered when rendering such services here in Gauteng province?What do you think can be done to improve access to health care services for refugees and asylum seekers here in Gauteng province?

#### 2.5.5. Data Analysis

According to Monette et al. [40], data analysis is the process of assembling and arranging information that has been collected from the participants and drawing meaning out of the large dataset that the researcher has. Data analysis assists the researcher to reduce the large batch of information into smaller bits of information, taking only what is relevant to the study and coming up with themes into which data can be classified [41]. The qualitative data analysis steps by Crewell [42] will be used to analyze the information gathered from the participants during the interviews.


Step 1: Organize and Prepare


Once the data have been collected from the participants, the interviews will be transcribed verbatim and the notes will be typed. The notes will then be arranged into various types based on the sources of information and also arranged into various types depending on the sources of information used during data collection;


Step 2: Reading through the data


After arranging the notes into different types, the researcher will read through them to gain the overall meaning of the participants’ responses. This process will help the researcher to have a general understanding/idea of what the participants are saying;


Step 3: Begin detailed analysis with a coding process


Having a general understanding of the data that have been collected from the participants will help the researcher to arrange the information into various categories. The categories will then be labeled with terms coming from the participants’ actual language or responses;


Step 4: Use the coding process to gather people and categories for analysis


When the data have been arranged into different categories, the themes that appear to be the major findings of the study will be generated. These themes will be reported under separate headings in the findings section of the study;


Step 5: Advance how the descriptive data and themes will be presented


The findings will be reported/conveyed in table form. The tables will be arranged according to the themes that the study will be presenting, and each will have its own categories;


Step 6: A finalized step in data analysis involves interpreting the data


The findings of the study will then be interpreted, explaining the lessons learned that capture the essence of the idea. The interpretation of the findings will be based on the researcher’s understanding of the phenomenon, with the integration of the relevant literature. The findings of study phase two will be used to guide the development of strategies to improve access to health care for refugees and asylum seekers.

#### 2.5.6. Trustworthiness

Trustworthiness will be ensured using the following four elements: (1) credibility, (2) conformability, (3) transferability, and (4) dependability. Each element will be explained in detail and the research will explain how the elements will be applied.

Credibility

Credibility refers to the context of ensuring that the information is being collected from relevant people who are informative about the phenomenon under investigation. The methods of ensuring credibility are prolonged engagement with the participants, persistent observation, and member checks [43]. Credibility will be ensured using prolonged engagement and member checks;

Prolonged engagement

The researcher will spend some time with the participants before the actual data collection process to build rapport and trust. This will help the participants to feel free to speak during the interviews. Furthermore, maintaining contact with the participants will help the researcher to ensure that they are relevant sources of information for the study;

Member checks

During the interview sessions, the researcher will probe more and paraphrase statements to have a clear understanding of what the participants are saying. A summary of the interview will be provided to all participants after each session and the tape recorder will be played back. This process will help the participants to add more information or to instruct the researcher to remove some of the information that was shared during the interview. If the need arises during the analysis stage/phase, the researcher will go back to the participants to seek clarity or additional information;

Conformability

The information recorded during the data collection process will be transcribed without any alterations. The researchers will be objective throughout the study, function as a research instrument, and not influence the outcome of the study or channel the participants into giving certain responses [43];

Transferability

The study approach, design, setting, target population, criteria of inclusion, sampling method, and procedure, as well as the theories used, will be fully described for generalizability. A full description of the study methodology will allow the study to be replicated by other researchers in the future and for them to arrive at a conclusion [43];

Dependability

Independent coders who are experts in the field of research will be allowed to co-code the data. The thesis will be submitted to external examiners and professionals will evaluate the strategies developed in phase two [43].

#### 2.5.7. Pre-Testing

Pre-testing will be conducted before the actual data collection process using participants who meet the inclusion criteria. Each participant will be selected from the sampled hospitals and metropolitan municipalities. The purpose of conducting pre-testing is to check if the questions are clear and direct or if there is a need to rephrase them [37].

### 2.6. Phase 1 (b): Quantitative Approach

This study will use a quantitative approach in phase 1 (b). This method is suitable for this section of the study due to its ability to analyze a large amount of data. Qualitative data will assist in strengthening the qualitative data that will be collected in phase 1 (a) of the study [38].

#### 2.6.1. Research Design

The study will adopt a cross-sectional descriptive design to strengthen the data collected through the qualitative approach. This design is considered to be relevant in this study since data will be analyzed at a specific point in time [44].

#### 2.6.2. Study Setting

The study will be conducted in Gauteng province in South Africa. Two metropolitan municipalities will be selected for this study, namely the City of Johannesburg and the City of Tshwane Metropolitan Municipality. From the two selected metropolitan municipalities, four hospitals will be selected, namely Kalafong Hospital, Steve Biko Academic Hospital, Charlotte Maxeke Academic Hospital, and Helen Joseph Hospital.

#### 2.6.3. Population and Sampling

The study population will consist of medical doctors and nurses who are responsible for providing health care services, as well as refugees and asylum seekers who have sought health services from any of the sampled/selected health facilities.

The respondents will be sampled through stratified random sampling. According to Creswell [42], stratified random sampling is appropriate for studies focusing on heterogeneous populations because it ensures the inclusion of small subgroups percentagewise. The population will be divided into different strata.

The researcher will request a list of refugees and asylum seekers who have sought medical attention from the four sampled hospitals in Johannesburg and Pretoria. Another list of medical doctors and professional nurses will be requested from Kalafong Hospital, Steve Biko Academic Hospital, Charlotte Maxeke Academic Hospital, and Helen Joseph Hospital. A total of 25% will be selected from each stratum for equal representation. The researcher will then apply Slovin’s formula to calculate the sample size. Each sample size will be increased by 10%, leaving room for non-responses. The sample size can only be determined after obtaining ethical clearance, since it is required for the researcher to access information from the relevant authorities.

#### 2.6.4. Data Collection

Two different questionnaires will be developed based on the findings from the qualitative approach, focusing on access to health care services for refugees and asylum seekers. One questionnaire will be for refugees and asylum seekers, whereas the other questionnaire will be for health practitioners. The questionnaires will be developed in English and translated into the native languages to cater to those who cannot read or understand English.

The researcher will ensure that appointments are made with the selected respondents for the distribution of the questionnaires. On the different dates agreed upon with the respondents, the researcher will then randomly distribute the questionnaires to the selected respondents who have agreed to form part of the study. Each questionnaire will take approximately 15 to 20 min to complete.

#### 2.6.5. Data Analysis

All completed questionnaires will be kept in a secure place where they will not be easily accessed by persons who are not part of the study. Data will be coded, captured, and cleaned before analysis. The questionnaires will be quantitatively analyzed by the researcher and a statistician on a computer using Statistical Package for Social Sciences (SPSS) version 26 for Windows. Descriptive statistics will be used to analyze the data. The chi squared test, which is known as a statistical test used to compare observed data and can only be used on actual numbers that are in the form of proportions, percentages, frequencies, and means, will be used. The analyzed data will be presented in graphs and charts [45].

#### 2.6.6. Ensuring Study Rigor

Validity

The promoter, co-promoters, and statistician will be awarded an opportunity to examine the instrument/questionnaire to ensure that its content is appropriate and to advise if there is any need for modifications to ensure that the instrument is aligned with the study objectives. Furthermore, the researcher will personally administer the questionnaires to the respondents to ensure validity [44];

Face validity

The instrument will be given to promoters who are experts in the field of study to check if it appears to be valid. Based on the feedback from the promoters and respondents, the questionnaire might be modified. The modified questionnaire will be administered to respondents who have shown interest and given consent to form part of the study [44];

Content validity

Those who are experts in the field of study will be allowed to assess the questionnaire to ensure content validity. A statistician will also be awarded an opportunity to go through the questionnaire for confirmation. This process will aid in improving the presentation of the content of the instrument [44];

Reliability

The research instrument/questionnaire will be pre-tested. The pre-testing of the questionnaire will be performed with 10% of the sample to check the reliability of the instrument. Once that process has been completed, the instrument will be modified/rectified and the researcher will re-administer it to the same respondents within two weeks. If the co-efficient reliability of the instrument is between 00 and 1.0%, the questionnaire will be sufficient to be used for the actual study because its reliability level will be high [46,47].

#### 2.6.7. Pre-Testing

The researcher will pre-test the questionnaire before the actual data collection process. The results emerging from the pre-testing of the instrument will be used to rectify and modify the questionnaire to ensure that it adequately measures the study variables. Ten percent of the total sample from different strata will be used for pre-testing. Respondents who will be selected for pre-testing will not form part of the main study. Once the instrument has been modified and rectified, it will be administered to the study respondents for data collection [37].

### 2.7. Phase 1 (c): Meta-Inferences and Conceptualization of the Findings

Anthony and Joseph [47] define meta-inference as an overall conclusion, description, or understanding developed through the incorporation of inferences gained from the qualitative and quantitative sections of a mixed-method study. The authors of [37] describe conceptualization as a method of breaking and altering research concepts into common meanings to develop an agreement among users. In this study, the overall conclusion will be drawn based on the results generated from both qualitative and quantitative approaches.

### 2.8. Phase 2: Development of Strategies

This phase of the study will focus on the development of strategies to address affordability, availability, accessibility, acceptability, and accommodation in order to improve access to health care services for refugees and asylum seekers. SWOT analysis and the build, overcome, explore and minimize (BOEM) model will be used to guide the process of strategy development. The Delphi technique will be used to validate the developed strategies.

SWOT analysis

SWOT analysis will be applied to the findings from the exploratory sequential mixed methods design in phase 1 of the study. This will be performed to identify the strengths that should be maintained or enhanced, the weaknesses that should be strengthened, the opportunities that can be exploited, and the threats that should be eliminated to improve access to health care services for refugees and asylum seekers;

BOEM model

After the SWOT analysis of the findings from phase 1 of the study, the BOEM model will be adopted. This model is best known for its ability to assist in developing strategies that overcome the weaknesses and threats of existing systems and to explore opportunities that can be used to achieve the overall goals. The strategies to be developed in this study will focus on overcoming threats and weaknesses to ensure affordability, availability, accessibility, acceptability, and accommodation in the current health system for refugees and asylum seekers. The researcher will ensure that strategies are developed in a manner that minimizes the chances of them failing to achieve their main goal, which is to improve access to health care services for refugees and asylum seekers.

### 2.9. Phase 3: Validation of Developed Strategies

This phase of the study will focus on the validation of the developed strategies to improve access to health care services for refugees and asylum seekers. The Delphi technique will be used to guide the validation of strategies.

Delphi Technique

The researcher will seek expert opinions on the developed strategies to improve access to health care services. This technique will assist in ensuring that the developed strategies are relevant to the desired goal. A total of 20 experts specializing in the field of study (including the promoter, co-promoters, and stakeholders) will be given an opportunity to critique the developed strategies based on their content and whether they will be able to ensure improved access to health care services for refugees and asylum seekers.

### 2.10. Ethical Principles

University protocols to ensure ethical principles

The study proposal was presented to the University of Venda Ethics Committee in a request for clearance; it was granted on the 25 July 2022 (FHS/22/PH/03/2507);

Permission to conduct the study

After being granted ethical clearance from the University of Venda, the proposal and ethical clearance certificate were submitted to the Gauteng Department of Health through the National Health Research Database portal to seek permission for the study. Gatekeeper has since granted permission for the study (GP_202207_088);

Informed consent

During the recruitment phase, participants will be informed about the aim of the study, its significance, and the method that will be used to collect data. The risks that participants are more likely to encounter/face during their participation in the study will be explained, if any. Furthermore, participants will be informed that they can still withdraw from the study at any time and that they will not be requested to give reasons for withdrawing their participation. Information sheets will be made available for those who can read to help them make informed decisions before giving consent. Those who are willing to be part of the study will be requested to give written or verbal consent [44];

Principle of Non-maleficence

The emotional, psychological, and physical well-being of the participants will be protected throughout the study. Sensitive words and questions that might negatively affect the well-being of the participants will be avoided when interacting with the participants. To protect participants from COVID-19 during the study, telephonic interviews will be conducted where possible. For those who will be meeting with the researcher face to face, social distancing will be maintained, sanitizers will be used, and masks provided to those who do not have them. Participants will not be requested to use their money or any other material resources for the benefit of the study [36];

Principle of Justice

All participants will be treated fairly throughout the study, they will not be discriminated against based on their socioeconomic status, educational level, age, ethnicity, religious beliefs, or any other factor. Furthermore, participation in the study will be voluntary [36];

Confidentiality

The information provided during the study will only be made available to persons who are involved in the study. The records made during the data collection process will be kept in a safe place where they cannot be accessed by those who are not part of the study [40];

Analysis and reporting

The information provided by the participants will not be altered to suit the aim and objectives of the study. All information will be reported verbatim. This principle will be complied with to ensure the trustworthiness of the study [36];

Anonymity

The actual names of the participants will not be used in the study. Each participant will be given a code, for example, “Participant 1”. The codes will be used when discussing the findings of the study. Furthermore, any information that could lead to the identification of participants will not be used in this study [44].

## 3. Results

Ethical clearance (FHS/22/PH/03/2507) and Gatekeeper permission (GP_202207_088) have been obtained from the relevant authorities. Refugees, asylum seekers, and health professionals will be recruited from the sampled hospitals in Gauteng province.

## Data Availability

Not applicable.

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
