# Peer review of "A Mixed Methods Protocol for Developing Strategies to Improve Access to Health Care Services for Refugees and Asylum Seekers in Gauteng Province, South Africa"

_healthcare, 2023, doi:10.3390/healthcare11172387_

Round 1

Reviewer 1 Report

I have pointed out the pdf. In-text references also need to be revised.

Grammatical errors are there. I have pointed them out in the pdf. 

Author Response

Responses have been added in the attached word document 

Reviewer 2 Report

First of all, this protocol draft can help - after a major revision - the authors design their planned research about an important topic indicated in the title. However, there are numerous issues to deal with. First and foremost, in an academic piece to get published in a fine journal any author must use fresh, updated data and literature, not just content from 2016, as it appears many times in this manuscript. Please, add more updated numbers, journal articles etc. When you write already in the first line of your abstract, "South Africa ... home to over 315 000 refugees and asylum seekers" ... where did you have this data from? Be as precise as possible throughout your paper!

It is, at the same time, definitely not true that this number makes South Africa the country in the continent with the highest cohort of such population! What about Uganda with the South Sudanese refugees, for instance, etc. When you make such statements, you have to back them firmly with data, literature, reports etc.

In 1.1. Definition of concepts: "Strategies" - is this really a concept? I do not think so! The language you use here (and throughout the paper) must be improved.

When you deal with the EU programme initiated in 2002, why not looking at the present-day situation, also in terms of policies, currently running programmes?

I would recommend that you deal more with Vision 2030 from all the perspectives of your planned research.

I think that the entire draft protocol is too fragmented, in certain instances, you could be more elaborate, in certain others, draw the different parts together. Already, in your abstract you should clearly say that this is a protocol type of piece.

Finally, I think that you also need to write about obstacles as well as the numerous challenges of the research.

I recommend a major revision.

For the literature review, several additional papers could be added, such as:

https://www.mdpi.com/2076-0760/12/5/284?type=check_update&version=2

https://journals.plos.org/plosone/article?id=10.1371/journal.pone.0271196

Musyimi C, Mutunga E, Ndetei D. COVID-19 and long-term care in Kenya. Country report available at LTCcovid.org, International Long-Term Care Policy Network, CPEC-LSE, 14 May 2020.

https://pubmed.ncbi.nlm.nih.gov/33282061/

There are several repetitions in the text. E.g. In 1. Introduction, already in the first sentence: "South Africa.... refugees and asylum seekers ..... to accommodate refugees and asylum seekers" Please, re-write in a more sophisticated manner linguistically.

In a number of cases, you use words with capital letters, such as Health practitioner, Refugees, etc..., which does not make sense. Please, correct.

In the first couple of paragraphs on Page 2, the tense you use is Present Tense, which is not appropriate.

I guess, in a scientific paper, you should avoid sentences such as the last one in the abstract.

Author Response

Comments have been addressed. Kindly see the attached word document 

Reviewer 3 Report

Thank you for the opportunity to review the work. It seems wise and presents valuable content; congratulations to the authors,

I have a few minor comments

- please put quotations at the end of the sentence, not in the middle or at the beginning

- Please start your sentence with a capital letter

- please use gender-neutral terms; instead of he/she, I suggest "subject."

- the project is described in detail; maybe there are too few summaries, and the reader gets lost. The inclusion of 40 participants seems to be a small research group.

- how do you plan to interview refugees and asylum seekers if the questionnaire is in English? How to ensure anonymity and privacy if refugees have to translate questions into their native language?

Please answer the questions. Regardless, I congratulate the idea and look forward to the results of the analyses.

no coments

Author Response

The responses have been addressed. See attached word document for point-by-point response

Round 2

Reviewer 1 Report

accepted

Reviewer 2 Report

The authors have dealt with the concerns I raised and answered all the questions I posed. I still think that the literature can be broadened. There are numerous minor spelling mistakes in the text, such as Campbel et al., [9], 62 Stated... there should be no capital S. Please, correct all of these!

It is fine, only some minor editing is needed.